# Molecular Analysis of Fetal and Adult Primary Human Liver Sinusoidal Endothelial Cells: A Comparison to Other Endothelial Cells

**DOI:** 10.3390/ijms21207776

**Published:** 2020-10-21

**Authors:** Muhammad Ahmer Jamil, Heike Singer, Rawya Al-Rifai, Nicole Nüsgen, Melanie Rath, Sascha Strauss, Ioanna Andreou, Johannes Oldenburg, Osman El-Maarri

**Affiliations:** 1Institute of Experimental Hematology and Transfusion Medicine, University of Bonn, Venusberg-Campus 1, 53127 Bonn, Germany; muhammad.jamil@ukbonn.de (M.A.J.); heike.singer@ukbonn.de (H.S.); rawya.al_rifai@ukbonn.de (R.A.-R.); nicole.nuesgen@ukbonn.de (N.N.); melanie.rath@ukbonn.de (M.R.); johannes.oldenburg@ukbonn.de (J.O.); 2Qiagen GmbH, 40724 Hilden, Germany; sascha.strauss@qiagen.com (S.S.); ioanna.andreou@qiagen.com (I.A.)

**Keywords:** LSECs, HHSEC, HCMEC, HPAEC, HPMEC, expression profiling, methylation profiling, genomic analysis, expression–methylation correlation, endothelial cells, F8 secreting cells

## Abstract

In humans, Factor VIII (F8) deficiency leads to hemophilia A and F8 is largely synthesized and secreted by the liver sinusoidal endothelial cells (LSECs). However, the specificity and characteristics of these cells in comparison to other endothelial cells is not well known. In this study, we performed genome wide expression and CpG methylation profiling of fetal and adult human primary LSECs together with other fetal primary endothelial cells from lung (micro-vascular and arterial), and heart (micro-vascular). Our results reveal expression and methylation markers distinguishing LSECs at both fetal and adult stages. Differential gene expression of fetal LSECs in comparison to other fetal endothelial cells pointed to several differentially regulated pathways and biofunctions in fetal LSECs. We used targeted bisulfite resequencing to confirm selected top differentially methylated regions. We further designed an assay where we used the selected methylation markers to test the degree of similarity of in-house iPS generated vascular endothelial cells to primary LSECs; a higher similarity was found to fetal than to adult LSECs. In this study, we provide a detailed molecular profile of LSECs and a guide to testing the effectiveness of production of in vitro differentiated LSECs.

## 1. Introduction

The F8 protein, whose deficiency leads to hemophilia A, plays an important role in blood coagulation [1]. The main cells contributing F8 to blood are the liver sinusoidal endothelial cells (LSECs) [2,3,4,5]. This has been proven in both mice and human cells [4,6]. Currently, there is no approved effective cure to treat F8 deficiency. Traditionally, the main treatment options were based on replacement therapies: either an on-demand or a prophylaxis-based intravenous injection of F8 protein. However, this requires frequent intravenous injections and is associated with high cost and, needless to say, will not provide a cure.

Current treatment options of F8 deficiencies include gene and cellular therapy. The former provides an actual option that could partially cure the disease as clinical trials show promising results [7]. Current gene therapy protocols rely on few available delivery tools and highly active non-factor 8 promoters (such as the tie2 promoter) that are actually needed and necessary to overcome the inefficient delivery tools and the low transfection efficiency. As a result, the low percentage of transfected cells are highly overexpressing F8 and thus could lead to high stress levels on the individual transfected cells and to an abnormal post translational modification of the F8 protein. The concept of cellular therapy, which is based on the use of functional therapeutic cells (i.e., LSECs, also known as human hepatic sinusoidal cells, HHSECs) offers an alternative to transferring only the F8 gene. The transfer of cells that are naturally producing F8 has been shown to be effective in treating hemophilia conditions, at least in mice [8,9,10], whereas the transplantation of fetal LSECs showed a 3-fold higher efficiency compared to adult LSECs in the repopulation of the liver in immunodeficient mice [11,12]. For this reason, we opted to study the molecular signature of the natural human cells producing F8 in both adult and fetal ECs, namely LSECs.

Previous studies addressed the genome wide molecular profiling (mRNA, microRNA or epigenetic markers) of endothelial cells (ECs) and/or LSECs in rodents [13,14,15,16] or in humans [17,18,19]. These early studies showed that different ECs can be classified based on their host organ [13,17], whether they are arterial or venous, and whether they are of micro- or macro vascular origin [17]. Recently, single cell RNA-seq for the whole liver resection has revealed molecular details for adult LSECs compared to other liver cell types [20,21,22].

In this study, we focused on comparing LSECs to other ECs and performed detailed molecular profiling of mRNA and CpGs methylation from both primary fetal and adult LSECs as well as other fetal ECs. Our detailed molecular analysis generated expression markers, methylation markers, and specific molecular/signaling pathways that distinguish fetal LSECs and adult LSECs from each other and from other studied fetal ECs. Our detailed molecular analyses lay the ground and establish the tools for comparison of in vitro generated ECs to primary ECs. Indeed, we generated iPS-derived vascular progenitor ECs (vECs) and found them to be closer to fetal than to adult LSECs.

## 2. Results

### 2.1. Comparison of Expression Profiles of Primary LSECs with Previously Published Single Cell LSECs RNA-Seq Data

Since our analyses required pure cell population, the obtained cells underwent an isolation procedure that included a purification step performed by the providers. Therefore, we verified the unchanged identity of the cells by comparing our data to available, previously published data from single cell liver analysis [21,22]. We isolated data corresponding to LSECs populations using bioinformatic techniques and compared the mean gene expression of the single cell LSECs with our primary LSECs data. Single cell data of LSECs and their differentially/highly expressed genes showed a significantly positive correlation for both adult (MacParland et al. [21] pearson correlation (Corr) = 0.5, *p* < 0.0005; Aizarani et al. [22] Corr = 0.51, *p* < 2.2 × 10 ^−16^ and fetal (MacParland et al. Corr = 0.23, *p* < 0.3; Aizarani et al. Corr = 0.50, *p* < 2.2 × 10 ^−16^) LSECs with periportal LSECs, whereas a negative trend and a non-significant correlation was found for central venous LSECs in both MacParland et al. and Aizarani et al. reported LSECs (Appendix A). The above comparisons clearly show that the primary cells used in this study shared LSECs identity markers. Additionally, we could also exclude gross culture effects as previous studies reported significant changes only after passages 3 to 4 [23,24,25].

### 2.2. FVIII Is Mainly Expressed in LSECs Followed by Other Endothelial Cells

F8 was reported as mainly expressed in LSECs as compared to hepatocytes and HUVECs (Human Umbilical Vein Endothelial Cells) [2], whereas another study showed that few other ECs can also produce F8 [4,26], therefore we compared the F8 expression in the following cells: human cardiac microvascular (HCMEC), human pulmonary arterial (HPAEC), and human pulmonary microvascular (HPMEC) ECs (collectively termed f-ECs) as well as in LSECs (adult and fetal). Expression levels confirmed that F8 is more expressed in LSECs (adult and fetal) followed by other fetal ECs (Appendix A). Overall F8 expression is very low in comparison to other genes but a fold change of 1.24 was observed between LSECs (adult and fetal) and other fetal ECs.

### 2.3. Comparison of LSECs with Other ECs Reveals Characteristics Detectable Already at the Studied First Trimester of Pregnancy

In order to investigate whether LSEC could be distinguished from other ECs at an early stage of development, we compared fetal LSECs (f-LSECs) to the fetal ECs. Differential gene expression analysis revealed 2547 probes (2304 genes) to be differentially expressed between f-LSECs and other f-ECs at *p* < 0.05 with 1395 probes (1243 genes) and 1152 probes (1069 genes) underexpressed and overexpressed, respectively, in f-LSECs in comparison to f-ECs (Figure 1A, data shown in Appendix A). Volcano plot reveals the differentially expressed genes based on both *p* value and mean differences (mean difference = fold change at log2 scale) (Appendix A). Similarly, differential methylation analysis identified 3223 differentially methylated CpGs (DMCs) between f-LSECs and f-ECs at FDR < 5% with 609 CpGs and 2614 CpGs hypomethylated and hypermethylated, respectively, in f-LSECs in comparison to f-ECs (Figure 1B). Further, we calculated the correlation between differentially expressed genes (DEGs) and DMCs (Figure 1C); this plot revealed excess of negative correlation when considering the promoter regions alone (58 vs. 25), but not at the non-promoter regions (44 vs. 56). The top 50 differentially expressed loci (sorted based on absolute mean difference) are shown in Figure 1D; the individual data of the top eight loci (based on absolute mean difference) are shown adjacent to the heatmap, including DNASE1L3, FCN3, H19, IGFBP5 and THY1 that are overexpressed in f-LSECs and EFEMP1, MGP, POSTN that are underexpressed. We also identified differentially methylated regions (DMRs) (present in TSS1500, TSS200, first exon and 5′UTR gene regions) in 96 genes (Figure 1E). DMR in GSTO2 were found to be the most significant. The individual data of the six most significant loci are shown adjacent to the heatmap in Figure 1E. In order to enable better insight into the biological functions and consequences of having differential gene expression between different ECs, we performed evidence/literature-based pathway analyses using IPA canonical pathways. The top affected canonical pathways include those related to protein synthesis and receptor signaling (where most affected genes are underexpressed in f-LSECs) (Figure 1F). Three pathways have more overexpressed genes in f-LSECs, and these include leukocyte extravasation signaling, which could be related to the known function of LSECs facilitating transport between blood stream and underlying liver cells [27]. The two remaining pathways are B cell receptor signaling and antigen presentation pathways, which are related to the known immunological function of LSECs [28,29,30]. Observing in more detail the pathway activation scores in Figure 1G, we found that only the EIF2 pathway was strongly inhibited in f-LSECs (Z score below 2 in all comparisons), whereas ATM signaling showed a borderline significant Z score, indicating activation in f-LSECs.

### 2.4. Comparison of Fetal LSEC vs. Adult LSEC Revealed Differences Related to Maturation and Cellular Specialization

Previous data have shown that cell identity is largely established during the early fetal stages. However, we would still expect differences related to maturation and further cellular specialization. Therefore, we compared the molecular profiling of adult vs. fetal LSECs. We identified differences in expression and methylation as follows: 1341 probes (of which 814 (762 genes) and 527 (505 genes) probes are over- (top genes: AKR1C3, BMP4, IFI27, MPZL2, POSTN and SLCO2A1) and underexpressed (top genes: MMP1, PHGDH) in a-LSECs, respectively), and differentially expressed between a-LSEC and f-LSEC at *p* < 0.05 (Figure 2A, data available in Appendix A), while DNA methylation showed 1056 DMCs (of which 413 and 643 CpGs were hyper and hypo methylated, respectively) at FDR < 5% (Figure 2B) (volcano plot for expersssion analysis in Appendix A). We found some bias (towards excess positive correlation) in the frequency of positive and negative correlations between DEGs and DMCs between a-LSECs and f-LSECs. In total, 22 were positively and 13 were negatively correlated when comparing between DEGs and DMCs, while, when considering only promoters, 11 were positively and seven were negatively correlated (Figure 2C). Using the single CpG methylation data, we identified DMRs (based on the protocol described in Materials and Methods) to distinguish adult from fetal LSECs. We found 54 DMRs to be differentially methylated when comparing between adult and fetal LSECs at promoter regions, whereby “MAB21L1” showed the most significant *p*-value at TSS200 (Figure 2E). The individual data of the six most significant loci are shown in Figure 2E adjacent to the heatmap (HOXD3, KRTCAP3, MAB21L1, MIR503, NBEA, and PCOLCE).

### 2.5. Whole Genome Bisulfite Sequencing Revealed More Epigenetic Markers that Illustrate the Progression from Fetal to LSEC

While Illumina arrays provide cost-effective and less technical variations, these arrays have the disadvantage of covering only a small percentage of genomic CpGs. The whole genome bisulfite sequencing (WGBS) method overcomes this low genomic coverage and has the potential to cover every CpG site in the genome that could have been missed or not covered by the arrays. We found no genome-wide significance between a-LSECs and f-LSECs (Figure 3A). At 5% FDR, we identified 273 DMCs (Figure 3B). Correlation between Illumina EPIC arrays and QIASeq Methyl WGBS showed an expected highly significant correlation of above 90% for both LSECs (adult and fetal) (Figure 3C). Additionally, based on our methodology to identify DMRs, we found 25 significant DMRs between a-LSEC and f-LSEC; six DMRs were selected as potent markers (Figure 3D) including regions in MGMT, CTSZ, and NACC2 genes.

### 2.6. Targeted Bisulfite Re-Sequencing Validates Selected Top Differentially Methylated Regions

We selected six regions from each of the three methylome-based comparisons (A. f-LSEC to other fetal (Figure 1E), B. f-LSEC to a-LSEC in arrays (Figure 2E), and C. f-LSEC to a-LSEC in WGBS (Figure 3D)) for verification by targeted bisulfite re-sequencing, whereby single DNA molecules are sequenced revealing the details of the methylation and the phase of epigenetic changes between neighboring CpGs. Using this approach, we were able to generate experimental data for 16 regions covering 258 CpGs. As a result of 3D-PCA analyses, the first group showed 26 CpGs in four regions (1:CDH5, 3:GSTO2, 5:RABGGTA and 6:ZC3H12D) with a clear separation between samples at ANOVA *p* < 0.05 (Figure 4, left part). Based on the second group, 98 CpGs in four regions (7:HOXD3, 9:MAB21L1, 10:MIR503 and 12:PCOLCE) clearly separated adult LSECs from other cell types (Figure 4, middle part). The third group revealed 53 CpGs in five regions (13:MGMT, 15:ASPSCR1, 16:CTSZ, 17: intergenic region, and 18:NACC2) and clearly separated a-LSEC from other samples (Figure 4, upper right part). As a result of the above analyses, we propose a panel from seven regions (3:GSTO2, 5:RABGGTA, 6:ZC3H12D, 7:HOXD3, 12:PCOLCE, 13:MGMT and 18:NACC2) that are particularly effective in identifying and distinguishing the adult and the fetal LSECs from each other and from other studied ECs (Figure 4, lower part).

### 2.7. Selected Methylation Panels Measure the Degree of Similarity between Primary and In Vitro Generated ECs

We generated patient specific iPS cells using blood from five different donors and differentiated them into progenitor vECs (Figure 5). The iPS cells showed markers of pluripotency, such as alkaline phosphatase, Nanog, SSEA-4, Tra-1-60 and Oct-4. In addition, they were able to form embryoid bodies and showed differentiation markers of all of the following embryonic layers: beta-tubulin III for ectoderm, actin smooth muscle for mesoderm, and alpha-fetoprotein for endoderm (Figure 5A). The iPS were differentiated into vECs according to a monolayer-based protocol using the small molecules CHIR and cytokines BMP4 and VEGFA [31]. The differentiated cells were stained for mesodermal/endothelial markers, such as brachyury, VE-cadherin, vWF, and CD31. They also supported tube formation and lipid uptake (Figure 5B). We used the panel of the 16 targeted bisulfites re-sequencing (as described in the previous section) to measure the degree of similarity of in-house generated ECs and primary cells. The results showed, for the in vitro differentiated progenitor vECs, that they are closer to non-LSECs; however, closer to fetal LSECs than to adult LSECs (Figure 4, lower part).

We also used the above described panel comparison tool to study the CpG methylation dynamic during the process of in vitro differentiation from blood cells to vECs—i.e., the initially used cells (PBMCs) in differentiation to the intermediate cells (EPC to iPS and to mesoderm) and ultimately to the final vECs (Figure 6). Based on unsupervised, as well as supervised (FDR < 5%), clustering analysis as illustrated by 3D-PCA and heatmap, we observed that PBMCs and EPCs were clustered close together as one group and that iPS cells, mesoderm and vECs were also clustered together (Figure 6, upper part). In fact, EPCs are derived from PBMCs and thus, in the studied methylation panel, they showed very high similarity rendering them almost indistinguishable from each other. This could be clearly observed at all regions as shown by the heatmaps (Figure 6, lower part). A similar scenario exists for the high degree of similarity between iPS, mesoderm and vECs, which still largely contained the methylation signals of the original iPS cells even after successful differentiation towards vECs. Such remaining signal is, for example, seen at the high methylation levels at regions 1 (CDH5), 2 (COL20A1), 6 (ZC3H12D), 13 (MGMT) and 18 (NACC2) and low methylation levels at region 10 (MIR503). However, from EPCs to iPS, there was a transition to another pattern of methylation shown by increase and decrease in methylation at regions 1 (CDH5), 2 (COL20A1), 6 (ZC3H12D), 13 (MGMT), 15 (ASPSCR1) and regions 4 (MIR21), 10 (MIR503), 12 (PCOLCE), respectively, possibly caused by the efficient reprograming of the cells induced by using transcription factors as mediators of reprogramming that were not used in the other steps (i.e., from PBMCs to EPCs and from iPS to mesoderm to vECs).

## 3. Discussion

Vascular endothelial cells are a layer of cells lining the inner walls of blood vessels/cavities. Although they are found all over the body and in different organs, they still share many essential biofunctions, such as angiogenesis, vascular development, and proliferation [13,17,32]. These biofunction similarities were also revealed in our study, where high similarities in expression as well as CpG methylation patterns were found. However, one of the main goals in the present study was to identify specific features in expression and methylation that distinguish LSECs from other ECs. This may help to better characterize these cells and may provide a link to their biofunctions (one of which is F8 production and secretion).

On the expression level, we observed a considerable number of DEGs (*p* < 0.05). Among the top ones are those that are closely linked to specific characteristics of LSECs, thus distinguishing them from other ECs. Despite the fact that all ECs share many similarities, LSECs also have specific expression signatures in comparison to other ECs. The top three LSECs over expressed genes are FCN3, IGFBP5 and DNASE1L3 (Figure 1D). Ficolin-3 (FCN3, also known as H-ficolin or Hakata antigen) is a pattern recognition molecule involved in the lectin-activating pathway of the complement system [33]. It circulates in blood and is synthesized mainly in the lung and the liver [34]. Akaiwa et al. could not detect the Ficolin-3 protein in LSECs, but found it in hepatocytes and, to a higher extent, in epithelial bile duct. Our data on the high expression of FCN3 in LSECs were also strongly confirmed by two further single cell sequencing studies by MacParland et al. and Aizarani et al. [21,22]. The second gene is IGFBP5, which codes for a secreted protein (the insulin-like growth factor-binding protein 5). It is a growth inhibiting factor that modulates and binds IGF1, downstream to mTOR1, inducing the expression of the hypoxia inducing factor 1 that provide positive feedback by directly activating the expression of IGFBP5 [35]. It also promotes the expression of pro-fibrotic and extra cellular matrix protein gene [36]. This suggests a less proliferative ability of LSECs in comparison to the other studied ECs and a local anti-proliferative effect on liver sinusoidal residual cells that may include hepatocytes. DNASE1L3 is a secreted deoxyribonuclease, known to be secreted by macrophages and dendritic cells, that digest serum DNA derived from apoptotic cells and neutrophils extracellular traps (NETs). In this way, DNASE1L3 will prevent an anti-DNA autoimmune response [37] and the possible vascular occlusion of micro vascular vessels by NETs [38].

When we considered the collective differential gene expression on specific canonical pathways we consistently found eight underactivated and three overactivated pathways (Figure 1G). These represent a bio-fingerprint that specifically distinguishes LSECs from other studied ECs and reflects their biofunction. The top underactivated pathway is EIF2 signaling, which is caused by a relative underexpression of the EIF2-beta subunit, suggesting a less active protein synthesis. The ATM signaling is the most consistently activated pathway due to lower expression of PP2A, resulting in a decrease in ATM dephosphorylation, leading to an increase in active ATM monomer. Moreover, several pathways related to the degradation of serotonin, dopamine and melatonin, are hypoactive due to a decrease in the expression of two main enzymes, namely aryl sulfotransferase and mono-amine oxidase. This leads to relatively higher levels of these products, specifically in LSECs and to a decrease in angiogenesis [39,40] and a decrease in pro-inflammatory cytokines [41].

In this study, we applied a DMR recognition strategy to identify reliable methylation markers that specifically identify LSECs. Both fetal and adult LSECs could be accurately identified and separated from each other, based on the identified methylation markers. The top DMR that distinguishes fetal LSECs from other fetal ECs is in the promoter area (TSS200) of the GSTO2 gene (*p* < 1 × 10 ^−5^), while the top DMR distinguishing between fetal and adult is in the promoter area of the MAB21L1 gene (*p* < 1.9 × 10 ^−7^). The underlying gene expression (for all top CpG markers) remained unaffected by the methylation differences. This may be due to the fact that the methylation levels are biological markers of the developmental history of the cells that were already established in earlier developmental stages [42]. However, the relative subtle methylation changes of up to 30% are possibly not sufficient to cause expression differences. Therefore, differential methylation markers could be a stable fingerprint, complementary to expression, to identify different ECs and their mature status of differentiation. For this purpose, we selected five and ten methylation DMRs markers that distinguish fetal LSECs from other fetal cells and fetal LSECs from adult cells (Figure 4) and performed bisulfite deep sequencing, confirming that the selected regions could effectively distinguish, in particular, adult LSECs. Next, we tested the distance relationship between the in vitro derived vECs and the primary LSECs and were able to localize the vECs closer to the precursor mesoderm cells than to any of the LSECs. Therefore, these methylation panels are useful markers to test the similarity of in vitro derived LSECs.

Some limitations of this study should be considered when interpreting the results and when building on them for future experiments. First: a larger study that includes more endothelial cells from multiple organs covering the entire human body is necessary to be performed in future. Second: our samples underwent a purification step to increase their purity and this was followed by two-dimensional cell culture that make them more robust and reliable for genomic analysis but not as native as single cell analysis from fresh tissues. [21,22] Third: a three-dimensional culture would have been closer to the native endothelial cells environment, as recent studies showed technical success in 3D cultures of brain endothelial cells to mimic the blood brain barrier. [43,44,45] However, a detailed transcriptome and methylome comparative analysis between 2D and 3D endothelial cells is still pending

In conclusion, in this study, we compared the molecular profile (expression and DNA methylation) of LSECs to selected vECs. We revealed several biological fingerprints of fetal LSECs, such as relatively higher expressions of FCN3, IGFBP5, DNASE1L3, THY1 and H19, but lower expressions of MGP, EFEMP1 and POSTN. Additionally, LSECs were shown to be characterized by the inhibition of EIF2 and mTOR signaling pathways. Finally, a panel of DMRs was identified in LSECs that can be used for the characterization of in vitro derived LSECs. Our study is the first to characterize LSECs in comparison to other vECs.

## 4. Materials and Methods

### Used Cellular Materials

Primary human endothelial cells: Commercial fetal (18 to 23 weeks of gestation) endothelial cells were supplied by ScienCell research laboratories (Corte Del Cedro, Carlsbad, CA, USA) via Provitro GmbH (Chariteplatz 1, Berlin, Germany) including LSECs (human hepatic sinusoidal endothelial cells), HPAEC (human pulmonary artery endothelial cells), HPMEC (human pulmonary microvascular endothelial cells), and HCMEC (human cardiac microvascular endothelial cells. Primary cells, DNA, small RNA and total RNA were available from three different male donors. All molecular materials (RNA and DNA from same donor) were isolated at first passage as stated by the manufacturing certificate. Adult LSECs material corresponding to two donors (males age 45 and 55 years old) was obtained from iXCells Biotechnologies (San Diego, CA, USA). All purities of cells were tested by flow cytometry and immunofluorescence staining by vendors; information is available on the respective vendor product page. Sciencell provides an ethical statement on isolation of the samples on their website: https://www.sciencellonline.com/ethical-statement. In addition, the ethics committee of the University Clinic Bonn approved the use of these samples (approval number: 041/13).

Generation of iPS cells: Peripheral blood mononuclear cells (PBMCs) from five donors were isolated from 2 × 10 mL peripheral blood using Fiquoll-Paque and subsequently expanded for 9–11 days for erythroid progenitor cells (EPCs) using the following cytokine cocktail: SCF 100 ng/mL, IGF1 40 ng/mL, IL3 10 ng/mL, EPO 2U/mL, Dexamethasone 1μM, (all in StemSpan SFEMII, Stemcell Technologies, Köln, Germany) (Ref: doi:10.3324/haematol.2009.019828). During conversion of PBMCs to EPCs primary cells were treated with Primocin & Pen/Strep to prevent microbial contamination (incl. Mycoplasma). For reprogramming 1 × 10^6^ EPCs were transferred by nucleofection using the Nucleofector 4D (LONZA, Köln, Germany) with a set of five episomal vectors (Epi5 Kit, Thermo Fisher, Darmstadt, Germany). For proof of pluripotency, iPS colonies were stained after 3–4 passages for alkaline phosphatase (AP Live Stain, Thermo Fisher, Darmstadt Germany) and immunostained for Oct4 (ab19857, Abcam, Berlin, Germany), Nanog (500-P236, Peprotech, Hamburg, Germany), Tra-1-60 (MAB4360, Merck Millipore, Darmstadt, Germany) and SSEA-4 (#60062, Stemcell Technologies, Köln, Germany). Cells were able to differentiate into all three germ layers: mesoderm (actin smooth muscle cell: MA5-11547, Thermo Fisher, Darmstadt, Germany), endoderm (alpha-fetoprotein: MAB1368, R&D Systems, Minneapolis, USA) and ectoderm (ß-tubulin III: #60100, Stemcell Technologies, Köln, Germany).

The generation of iPS-derived vECs: Differentiation from iPS into vECs was performed according to a published protocol [31]. All differentiations were performed using Mycoplasma-free IPS clones at low passage (pass 5–7) to ensure genomic integrity measured with corresponding IPS clones at pass 3–4. Mycoplasma testing was performed regularly using the DAPI staining method. Briefly, after generating an iPS single cell suspension, cells were reseeded on matrigel in mTeSR1 containing rock inhibitor. After one day, medium was switched to N2B27 medium containing CHIR99021 (8 μM/mL) and BMP4 (25 ng/mL). After three days, medium was changed to vEC induction medium (StemPro34 containing 200 ng/mL VEGF-A and 2 μM/mL forskolin), directing the differentiation from mesoderm into generic vascular endothelial cells. On day 6, vECs were MACS separated using CD144 MicroBeads (Miltenyi Biotec, Bergisch Gladbach, Germany). Positive sorted cells were replated on 0.2 μg/cm^2^ human fibronectin (Sigma Aldrich, Taufkirchen, Germany) and further cultured in EGM-2 (#CC-3162, LONZA, Köln, Germany). The efficiency of differentiation was analyzed by immunofluorescent (IF) staining and SYBR green rtPCR (data not shown). For IF staining, cells were stained according to the standard method, with the specific mesodermal marker anti-brachyury (AF2085, R&D Systems, Minneapolis, USA) and the endothelial markers anti-VE-cadherin (sc-9989, Santa Cruz, Heidelberg, Germany), anti-CD31 (sc-1506, Santa Cruz, Heidelberg, Germany) and anti-vWF (A0082, Dako, Bath, UK). For SYBR, green rtPCR, RNA was isolated on day 0, day 3 and day 6 during differentiation. The relative gene expression of the pluripotency markers Oct4 and Sox2 decreased over time, while the specific mesoderm markers brachyury and MIXL increased only on day 3. Endothelial cell markers CD31 and CD34 only appeared on day 6 of differentiation. For proof of angiogenic potential, 7 × 104 MACS-isolated vECs were seeded on a 24-well plate coated with 300 uL growth-reduced factor matrigel (#356230, Corning) in EGM-2 medium (LONZA, Köln, Germany). The ability to form tubular structures was investigated periodically after 2 h, 4 h, 6 h and 8 h on a light microscope (Zeiss, Oberkochen, Germany). For lipid uptake assay, 3 × 10^4^ vECs were seeded in triplicate on a 96-well plate with EGM-2 medium. After 48 h, medium was supplemented with 1:100 diluted LDL-DyLightTM 550 (#ab133127, Abcam, Berlin, Germany). After 5 h, the degree of LDL uptake was examined under a fluorescent microscope (Axio Observer.7 with ApoTome.2, Zeiss, Oberkochen, Germany). Cells were also verified for genomic integrity using Illumina psych array 2.0 (data not shown). Cellular materials of DNA were obtained from differentiated cells. The starting material of PBMCs (peripheral blood mononuclear cells) and the intermediate derived EPC (erythroid progenitor cells), iPSCs (induced pluripotent stem cells) and mesoderm, together with the ultimately day 6 vEC (vascular endothelial cells) were available from five different blood donors. DNA was extracted using the Qiagen “MiniPrep” kit.

Single-Cell Data Analysis: Single cell data of liver were downloaded from NCBI GEO with an accession number of “GSE124396” (Aizarani et al.) [22] and “GSE115469” (Marpaland et al.) [21]. Macparland et al. data were analyzed using Seurat [46] package in R and central venous LSECs, periportal LSECs and portal ECs genes, and their respective expression values were extracted from the table. Aizarani et al. data were downloaded from their supplementary data repositories for LSEC expressed genes and k-means clustering was performed on the genes with k = 3. Genes were divided into A, B and C clusters for further analysis. Correlation was performed based on differentially expressed genes from macparland et al. and LSECs expressed genes were used to calculate correlation between Aizarani et al. and arrays data.

Arrays analyses: All expression (Illumina Human HT12-V4) and methylation data (Illumina 450K methylation, and Illumina EPIC methylation) were analyzed using packages available in R. Quality control was performed for all arrays and all samples passed the quality control test (Data not shown; raw data available on NCBI GEO). Illumina 450K methylation arrays were used for fetal LSECs and other fetal ECs, whereas a latest version Illumina EPIC methylation was used for adult and fetal LSECs comparison. Differentially expressed genes and differentially methylated CpGs were identified using Qlucore Omics Explorer 3.5 (Qlucore AB, Lund, Sweden), while an improvised methodology (details in Appendix B) was used to identify differentially methylated regions.

Next generation sequencing data analysis: Whole genome bisulfite sequencing (QIAseq Methyl Library Kit) and targeted next-generation sequencing methylation (QIAseq Targeted Methyl Panel Kit) data were obtained and analyzed using a pipeline (described in Appendix C). A novel method was used to identify the DMRs in whole genome bisulfite sequencing data (details in Appendix D). Targeted NGS methylation was chosen to confirm the methylation differences at selected DMRs.

Pathway analyses: Canonical pathway enrichment was performed using Ingenuity Pathway Analysis (IPA). IPA has detailed canonical pathways from specific journal articles, review articles, text books and HumanCyc (IPA Tutorials). Differentially expressed genes were entered into IPA and mean expression difference was used as expression log ratio to calculate activation and inhibition of the pathway. Activity score was calculated using Z score by IPA [47]. A Z score above 0 was considered activated while a Z score below 0 was considered inhibited.

Statistical analyses: Anova test was performed between multiple samples and student *t*-test was performed between two sample analyses. *p* values were adjusted using the Benjamini and Hochberg method (FDR) [48]. Fold changes alone ignore the variability of the data and assumes equal variance for all genes across the array [49]; therefore, statistically, *p* < 0.05 or 5% of FDR was considered significant whereas top differentially expressed genes were selected based on high absolute mean differences (mean difference = fold change with log2 scale).

## Figures and Tables

**Figure 1 ijms-21-07776-f001:**
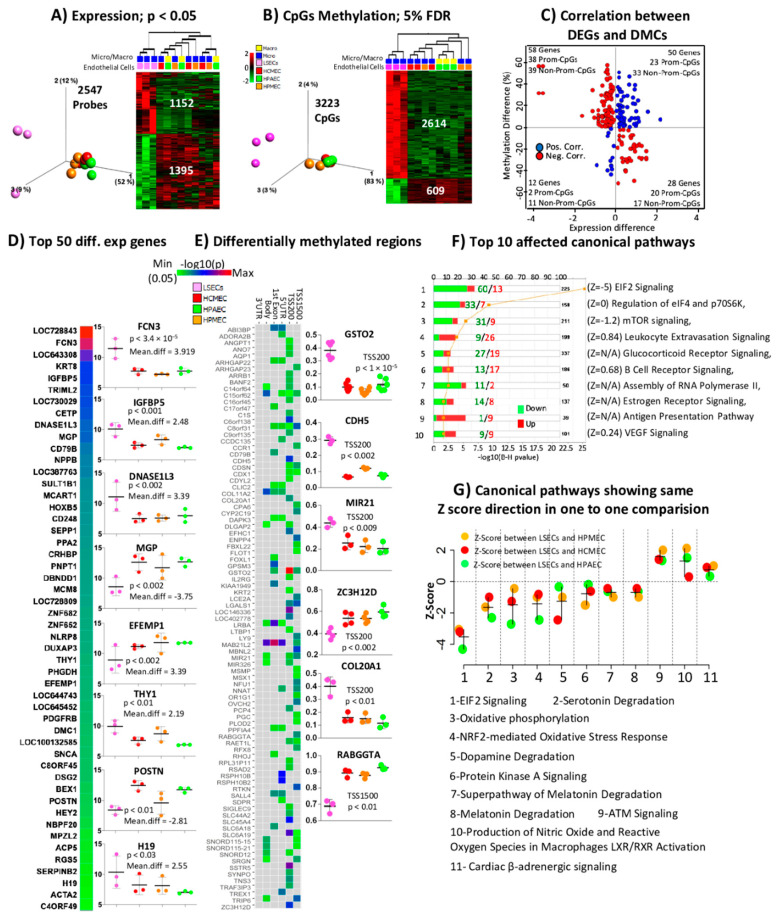
Expression and CpGs methylation profiling in fetal liver sinusoidal endothelial cells (f-LSECs) and other fetal endothelial. Cells (f-ECs): Human cardiac microvascular endothelial cells (HCMEC), Human pulmonary arterial endothelial cells (HPAEC) and Human pulmonary microvascular endothelial cells (HPMEC). Three-dimonsional-PCA plot and heatmap of (**A**) differentially expressed genes (DEGs) (at *p*< 0.05) and (**B**) differentially methylated CpGs (DMCs) (at 5% of FDR). (**C**) Correlation (Pearson > 0.7) between DEGs (at *p* < 0.05) and CpG methylation (red and blue correspond to negative and positive correlation, respectively). (**D**) Heatmap of −log10(p) and mean expression difference (fold change in log2 scale) of top 50 DEGs between f-LSEC and f-EC (individual data of top eight DEGs, based on absolute mean expression difference, are shown in box plot). (**E**) Heatmap of all differentially methylated regions comparing f-LSECs and other f-ECs. (Individual data of best six loci are shown in box plot.) (**F**) Top ten affected canonical pathways enriched with DEGs compared between f-LSECs and other f-ECs with a number of upregulated and downregulated genes. (**G**) Selected differentially affected canonical pathways identified by IPA in comparative analysis of f-LSECs and other f-ECs. Only those displaying the same comparative directions of Z score are shown (i.e., all Z scores negative or positive in comparison to f-LSECs).

**Figure 2 ijms-21-07776-f002:**
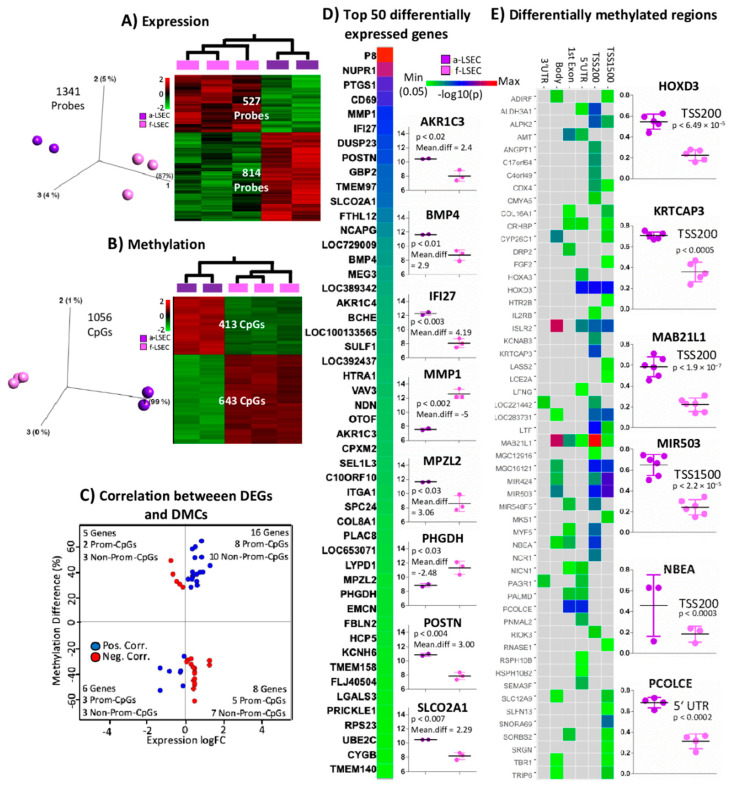
Expression and CpGs methylation profiling in adult and fetal liver sinusoidal endothelial cells (a-LSECs, f-LSECs). 3D-PCA plot and heatmap of (**A**) Differentially expressed genes (DEGs) (at *p* < 0.05) and (**B**) Differentially methylated CpGs (DMCs) (at 5% of FDR and 0.3 projection score) between a-LSECs and f-LSECs. (**C**) Correlation (Pearson > 0.7) between DEGs (at *p* < 0.05) and CpG methylation (red and blue correspond to negative and positive correlation, respectively). (**D**) Heatmap of −log10(p) and mean difference (fold change in log2 scale) for top 50 DEGs between a-LSECs and f-LSECs (box plots represent the top eight significant loci based on absolute mean expression difference). (**E**) Heatmap of differentially methylated regions (DMRs) comparison between a-LSECs and f-LSECs with respect to different regions of a gene (i.e., TSS1500, TSS200, 5′UTR, 1stExon, body, 3′UTR) (box plots represent the six most significant markers).

**Figure 3 ijms-21-07776-f003:**
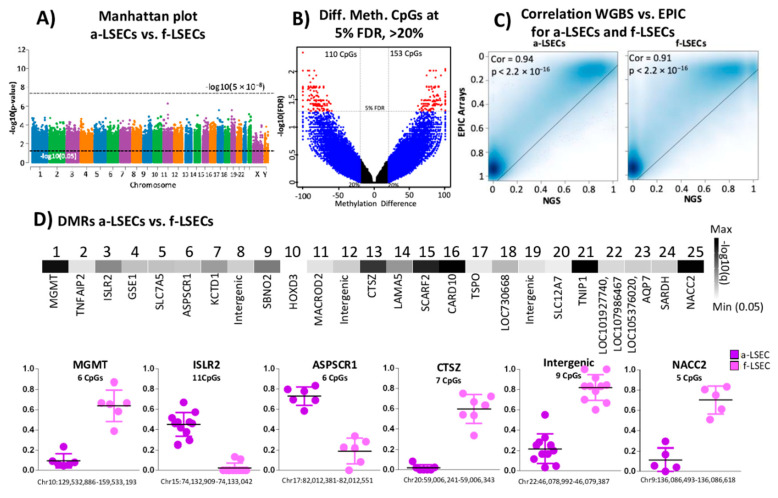
Whole genome bisulfite sequencing data analysis between adult and fetal liver sinusoidal endothelial cells (a-LSECs, f-LSECs). (**A**) Genome-wide Manhattan plot of –log10(p) between a-LSECs and f-LSECs. (**B**) Volcano plot of methylation difference and –log10(FDR) between a-LSECs and f-LSECs. (**C**) Correlation and density plot between Illumina EPIC methylation and QIASeq methyl whole genome bisulfite sequencing data in adult and fetal LSECs. (**D**) Heatmap of significant differentially methylated regions (DMRs) between a-LSECs and f-LSECs. Below box plots show the individual CpGs methylation of a-LSECs and f-LSECs in selected DMRs.

**Figure 4 ijms-21-07776-f004:**
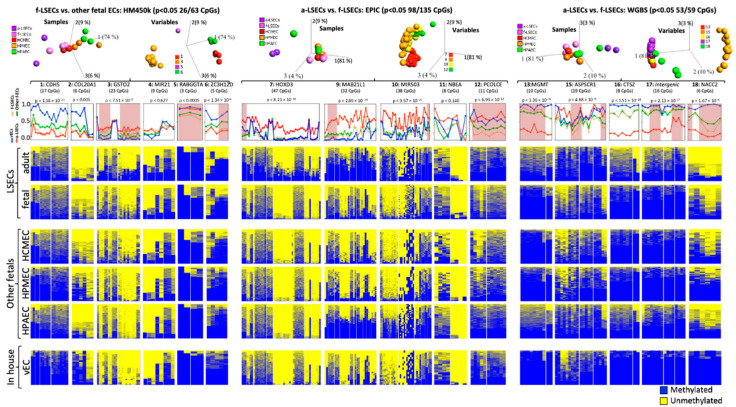
Targeted bisulfite re-sequencing in different endothelial cells. Analysis of statistical significance of 16 selected regions comparing adult and fetal liver sinusoidal endothelial cells (a-LSECs and f-LSECs) and other fetal endothelial cells (f-ECs). Top panel: PCA plot of samples and variable for the selected regions at *p* < 0.05. Below is the name of the gene host of the differentially methylated region (DMR), followed by line plots representing the average methylation of all CpGs at a region in a-LSECs, f-LSECs, f-ECs and vascular endothelial cells (vECs) (the regions that have the power to clearly distinguish between a-LSECs, f-LSECs and other f-ECs are highlighted in orange). In the bottom part are heatmaps (of all samples combined) of every region for methylation patterns in a-LSECs, f-LSECs, f-ECs (human cardiac microvascular endothelial cells (HCMEC), human pulmonary microvascular endothelial cells (HPMEC), and human pulmonary arterial endothelial cells (HPAEC)) and vECs.

**Figure 5 ijms-21-07776-f005:**
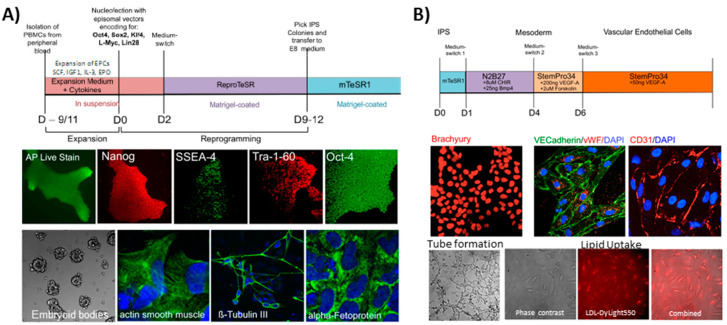
Characterization of iPS and generated endothelial cells. (**A**) The upper part shows the detailed program used to generate iPS cells from PBMCs. Middle panel shows histo-immunostaining with pluripotency markers: alkaline phosphatase, Nanog, SSEA-4, Tra-1-60 and Oct-4. The lower panel shows the potential of the iPS cells to form embryoid bodies and to differentiate in all three germ layers: mesoderm (actin smooth muscle cell), endoderm (alpha-fetoprotein) and ectoderm (ß-tubulin III). (**B**) The upper part shows the detailed protocol used to derive iPS-endothelial cells. The middle part shows the histo-immunostaining with typical endothelial markers: brachyury, VE-cadherin, vWF and CD31. The lower panel shows the results of endothelial cell assay characterization of tube formation and lipid uptake.

**Figure 6 ijms-21-07776-f006:**
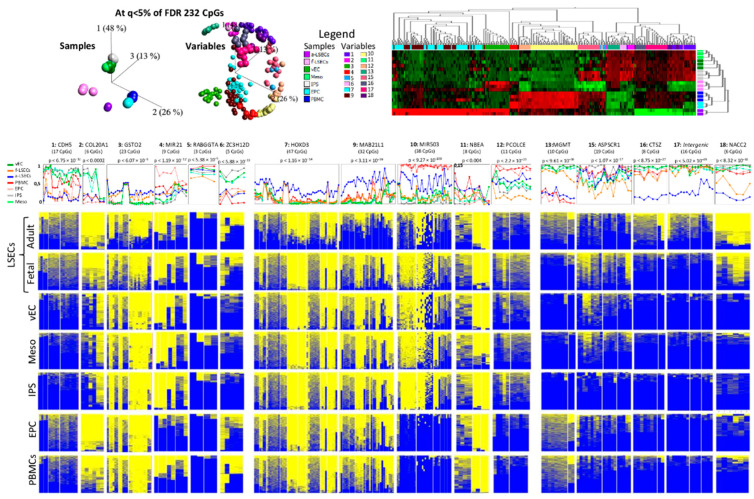
Targeted bisulfite sequencing in different endothelial cells and its precursors. Analysis of statistical significance of 16 selected regions between adult and fetal liver sinusoidal endothelial cells (a-LSECs and f-LSECs), vascular endothelial cells (vECs), mesoderm derived cells, induced pluripotent cells (iPS), erythroid progenitor cells (EPCs) and peripheral blood mononuclear cells (PBMCs). Upper panel: PCA and heatmap for significant CpGs using Anova at 5% of FDR between a-LSECs, f-LSECs, PBMCs, EPCs, iPSc, mesoderm, and vECs. Next, the line plot for mean average per CpG for every region, followed by heatmap for every region showing methylation pattern in all samples.

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
