# Peer review of "Molecular Analysis of Fetal and Adult Primary Human Liver Sinusoidal Endothelial Cells: A Comparison to Other Endothelial Cells"

_ijms, 2020, doi:10.3390/ijms21207776_

Round 1
Reviewer 1 Report
Interesting and well written paper with obvious translational implications.
More details on the iPS cell derivation are needed (pag. 11, line 348-350): "subsequently expanded for 9-11 days for erythroid progenitor cells (EPCs)".
You have to be more specific to make your procedure repeatable and reproducible.
Minor point: ref 7 has to be amended.
Author Response
Interesting and well written paper with obvious translational implications.
More details on the iPS cell derivation are needed (pag. 11, line 348-350): "subsequently expanded for 9-11 days for erythroid progenitor cells (EPCs)".
You have to be more specific to make your procedure repeatable and reproducible.
Response: Details about iPS cell derivation has been added in method section.
“Peripheral blood mononuclear cells (PBMCs) from five donors were isolated from 2x10ml peripheral blood using Fiquoll-Paque and subsequently expanded for 9-11 days for erythroid progenitor cells (EPCs) using the following cytokine-cocktail: SCF 100ng/ml, IGF1 40ng/ml, IL3 10ng/ml, EPO 2U/ml, Dexamethasone 1μM, (all in StemSpan SFEMII, Stemcell Technologies) (Ref: doi: 10.3324/haematol.2009.019828). During conversion of PBMCs to EPCs primary cells were treated with Primocin & Pen/Strep to prevent microbial contamination (incl Mycoplasma).”
Minor point: ref 7 has to be amended.
Response: Ref.7 has been corrected.
Peyvandi F, Garagiola I. Clinical advances in gene therapy updates on clinical trials of gene therapy in haemophilia. Haemophilia. 2019 Sep;25(5):738-46.
Reviewer 2 Report
Dear Editor
The study by Jamil and Singer et al. provides a genetic insight regarding the characterization of liver sinusoidal endothelial cells in comparison to other iPS-derived vascular progenitor endothelial cells.
Moreover, it reports a detailed molecular profiling of mRNA and CpGs methylation from both primary fetal and adult LSECs as well as other fetal ECs.
The design of the study and the technical quality of the work are convincing and results can be of general interest. The manuscript is well-written and easy to follow. Moreover, authors have successfully managed to discuss the major findings of their study through an unbiased comparison with a good range of up-to-date literature.
However, there is a number of points that would need to be addressed in order to improve the quality of this paper before it can be accepted for publication:
-Limited passage number is one of the major problems with iPS cells. Authors needs to mention the utilised passage number (range) and include a statement about their measures for mycoplasma testing.
-In data analysis line 101, authors only provide the p value without determining a fold change cut-off value. FC values of at least 1.3 is a common practice in this type of analysis.
-The study is based on transcriptomic profiling of cells which are grown in 2D. New lines of research have discussed the variability of genetic and protein profiling in 2D vs 3D. Authors need to discuss this as a limitation of the current study. Future studies can benefit from 3D self-organized microvascular models and microvessel-on-a-chip platforms. References to be included:
- https://pubmed.ncbi.nlm.nih.gov/30032046/
- https://www.frontiersin.org/articles/10.3389/fbioe.2020.573775/full
- https://pubmed.ncbi.nlm.nih.gov/30165870/
-Line 118, authors discussed the identification of three suggested pathways. What was the platforms for data analysis and has KEGG been considered for data analysis?
Best.
Author Response
The study by Jamil and Singer et al. provides a genetic insight regarding the characterization of liver sinusoidal endothelial cells in comparison to other iPS-derived vascular progenitor endothelial cells.
Moreover, it reports a detailed molecular profiling of mRNA and CpGs methylation from both primary fetal and adult LSECs as well as other fetal ECs.
The design of the study and the technical quality of the work are convincing and results can be of general interest. The manuscript is well-written and easy to follow. Moreover, authors have successfully managed to discuss the major findings of their study through an unbiased comparison with a good range of up-to-date literature.
However, there is a number of points that would need to be addressed in order to improve the quality of this paper before it can be accepted for publication:
Response: We would like to thanks reviewer for their thoughtful comments which are really beneficial and will improve our manuscript.
-Limited passage number is one of the major problems with iPS cells. Authors needs to mention the utilised passage number (range) and include a statement about their measures for mycoplasma testing.
Response: We have added the following information in method section “Generation of iPS-derived vECs”.
“All differentiations were performed using Mycoplasma-free IPS clones at low passage (pass 5-7) to ensure genomic integrity measured with corresponding IPS clones at pass 3-4. Mycoplasma-testing was performed regularly using the DAPI staining method.”
-In data analysis line 101, authors only provide the p value without determining a fold change cut-off value. FC values of at least 1.3 is a common practice in this type of analysis.
Response: Thanks reviewer for pointing out the missing information about fold change. Fold changes are known cut-off to identify differentially expressed genes so we first consider the p-value as threshold and then identify the best 50 genes considering the highest absolute mean difference (mean difference is the fold change in log2 scale).
We have added the detail gene selection methodology in method section and also a supplementary figure 3 and 4 showing a volcano plot showing subset of genes with p-value as well as mean difference (log2 foldchange).
“Fold changes alone ignore the variability of the data and assumes equal variance for all genes across the array [48] therefore statistically, p<0.05 or 5% of FDR was considered significant whereas top differentially expressed genes were selected based on high absolute mean differences (mean difference = fold change with log2 scale)”
Chen, J., Wang, S., Tsai, C. et al. Selection of differentially expressed genes in microarray data analysis. Pharmacogenomics J 7, 212–220 (2007).
-The study is based on transcriptomic profiling of cells which are grown in 2D. New lines of research have discussed the variability of genetic and protein profiling in 2D vs 3D. Authors need to discuss this as a limitation of the current study. Future studies can benefit from 3D self-organized microvascular models and microvessel-on-a-chip platforms. References to be included:
- https://pubmed.ncbi.nlm.nih.gov/30032046/
- https://www.frontiersin.org/articles/10.3389/fbioe.2020.573775/full
- https://pubmed.ncbi.nlm.nih.gov/30165870/
Response: Thanks for the advice, we are planning this as our future prospect and will work in future for the genetic as well as protein profiling of LSECs.
- Line 118, authors discussed the identification of three suggested pathways. What was the platforms for data analysis and has KEGG been considered for data analysis?
Response: Platform used for data enrichment analysis is Ingenuity pathway analysis. Details mentioned in method section under pathway analysis heading also added in results that pathways analysis was performed using IPA. KEGG pathways are included in HumanCyc, and IPA considered canonical pathways from articles, text books, journals and HumanCyc.
“Canonical pathways enrichment was performed using Ingenuity Pathway Analysis (IPA). IPA has detailed canonical pathways from specific journal articles, review articles, text books and HumanCyc (IPA Tutorials).”
Round 2
Reviewer 2 Report
Dear Editor
I would like to thank the authors for their efforts to revise the manuscript in the light of the raised concerns and suggestions. The newly added details have helped towards the improvement of the current version compared to their earlier submission.
The majority of my comments have been addressed by the authors accordingly but they missed to address the limitation about 2D vs 3D models which should be added at the end of discussion.
I would like to recommend this manuscript for publication at IJMS once the below mentioned point is correctly addressed.
Best.
-The study is based on transcriptomic profiling of cells which are grown in 2D. New lines of research have discussed the variability of genetic and protein profiling in 2D vs 3D. Authors need to discuss this as a limitation of the current study. Future studies can benefit from 3D self-organized microvascular models and microvessel-on-a-chip platforms. References to be included:
- https://pubmed.ncbi.nlm.nih.gov/30032046/
- https://www.frontiersin.org/articles/10.3389/fbioe.2020.573775/full
- https://pubmed.ncbi.nlm.nih.gov/30165870/
Author Response
I would like to thank the authors for their efforts to revise the manuscript in the light of the raised concerns and suggestions. The newly added details have helped towards the improvement of the current version compared to their earlier submission.
The majority of my comments have been addressed by the authors accordingly but they missed to address the limitation about 2D vs 3D models which should be added at the end of discussion.
I would like to recommend this manuscript for publication at IJMS once the below mentioned point is correctly addressed.
Best.
-The study is based on transcriptomic profiling of cells which are grown in 2D. New lines of research have discussed the variability of genetic and protein profiling in 2D vs 3D. Authors need to discuss this as a limitation of the current study. Future studies can benefit from 3D self-organized microvascular models and microvessel-on-a-chip platforms. References to be included:
- https://pubmed.ncbi.nlm.nih.gov/30032046/
- https://www.frontiersin.org/articles/10.3389/fbioe.2020.573775/full
- https://pubmed.ncbi.nlm.nih.gov/30165870/
Response: We would like to thank reviewers for their kind remarks. We have added the relevant information regarding limitation of our study, including the references mentioned by reviewer.
“Some limitations of this study should be considered when interpreting the results and when building on them for future experiments. First: a larger study that include more endothelial cells from multiple organs covering the entire human body is necessary to be performed in future. Second: our samples underwent purification step to increase their purity and this was followed by 2-dimetional cell culture that make them more robust and reliable for genomic analysis but not as native as single cell analysis from fresh tissues.[21, 22] Third: a three-dimensional culture would have been closer to the native endothelial cells environment as recent studies showed technical success in 3D cultures of brain endothelial cells to mimic blood brain barrier.[45-47] However a detailed transcriptome and methylome comparative analysis between 2D and 3D endothelial cells is still pending”